# Novel Endocrine Therapeutic Opportunities for Estrogen Receptor-Positive Ovarian Cancer—What Can We Learn from Breast Cancer?

**DOI:** 10.3390/cancers16101862

**Published:** 2024-05-13

**Authors:** Tine Ottenbourgs, Els Van Nieuwenhuysen

**Affiliations:** 1Gynaecological Oncology Laboratory, KU Leuven, Leuven Cancer Institute, 3000 Leuven, Belgium; tine.ottenbourgs@kuleuven.be; 2Department of Gynaecology and Obstetrics, University Hospitals Leuven, BGOG and Leuven Cancer Institute, 3000 Leuven, Belgium

**Keywords:** low-grade serous ovarian cancer, breast cancer, estrogen receptor, endocrine resistance, combination therapy, novel endocrine therapies

## Abstract

**Simple Summary:**

Low-grade serous ovarian cancer is a rare type of ovarian cancer that usually has an indolent growth and affects younger women. It has markers that suggest it might respond to hormone therapy, but unfortunately, this treatment does not work well for many patients because the cancer becomes resistant to it. We do not fully understand why this happens. In breast cancer, similar resistance mechanisms are studied, so we are exploring if we can apply what we learn there to improve treatment for this type of ovarian cancer. This review looks at why hormone therapy might stop working in ovarian cancer and explores new ways to make it more effective. The goal is to find better treatment options for patients with advanced low-grade serous ovarian cancer, who currently do not have many choices for treatment.

**Abstract:**

Low-grade serous ovarian cancer (LGSOC) is a rare ovarian malignancy primarily affecting younger women and is characterized by an indolent growth pattern. It exhibits indolent growth and high estrogen/progesterone receptor expression, suggesting potential responsiveness to endocrine therapy. However, treatment efficacy remains limited due to the development of endocrine resistance. The mechanisms of resistance, whether primary or acquired, are still largely unknown and present a significant hurdle in achieving favorable treatment outcomes with endocrine therapy in these patients. In estrogen receptor-positive breast cancer, mechanisms of endocrine resistance have been largely explored and novel treatment strategies to overcome resistance have emerged. Considering the shared estrogen receptor positivity in LGSOC and breast cancer, we wanted to explore whether there are any parallel mechanisms of resistance and whether we can extend endocrine breast cancer treatments to LGSOC. This review aims to highlight the underlying molecular mechanisms possibly driving endocrine resistance in ovarian cancer, while also exploring the available therapeutic opportunities to overcome this resistance. By unraveling the potential pathways involved and examining emerging strategies, this review explores valuable insights for advancing treatment options and improving patient outcomes in LGSOC, which has limited therapeutic options available.

## 1. Introduction

Low-grade serous ovarian cancer (LGSOC) is a rare histologic subtype of epithelial ovarian carcinoma, accounting for approximately 10% of all ovarian cancer cases [1]. It is clinically, histologically, and molecularly distinct from the most common type of ovarian cancer, high-grade serous ovarian cancer (HGSOC). Generally, women with LGSOC are diagnosed at a younger age (43–55 years) and have a prolonged overall survival [1]. LGSOC is often characterized by high estrogen and progesterone receptor positivity. Additionally, they usually show an activated mitogen-activated protein kinase (MAPK) pathway with *KRAS* and *BRAF* mutations and, unlike patients with HGSOC, demonstrate a wild-type TP53 expression pattern [2].

LGSOC has a good prognosis in the early stages of the disease, but treating advanced and recurrent cases of the disease poses significant challenges. The current treatment approach to LGSOC primarily relies on the treatment strategies of HGSOC, involving debulking surgery and platinum-based chemotherapy, even though LGSOC is considered to be potentially resistant to chemotherapeutic agents. In recurrent cases, reported overall response rates (ORRs) to chemotherapy range from 2.1% to 17% [3,4]. These numbers emphasize that there is a need for the development of improved, more tailored treatment strategies, especially in the recurrent setting, where no standard-of-care treatment is available [5]. These treatment strategies should be focused on the molecular characteristics of LGSOC in order to improve clinical outcomes.

Numerous studies have emphasized the similarities between LGSOC and estrogen receptor-positive (ER+) breast cancer. The first similarity is the expression of the estrogen receptor (ER). The majority of LGSOCs have a strong ER positivity, and most tumors also demonstrate positivity for the progesterone receptor [6,7]. Additionally, both cancer types have a relatively indolent disease course, compared with their hormone receptor-negative or high-grade counterparts. They may progress slowly over time, allowing for long-term management strategies [6].

In early-stage ER+ breast cancer, endocrine therapy has led to significant reductions in both recurrence and mortality rates [8]. However, in the metastatic and recurrent setting, resistance to treatment and disease progression inevitably occurs, despite initial responses to endocrine therapy [9]. The response to endocrine therapy may be influenced by acquired or intrinsic factors that contribute to endocrine resistance. In metastatic or advanced breast cancer, acquired resistance typically emerges after an initial response to therapy, which generally occurs after six or more months of treatment. In contrast, metastatic or advanced breast cancers that are intrinsically resistant may not respond to treatment within a shorter timeframe, typically less than 6 months [10].

According to published data, the level of ER expression has been identified as the most reliable predictor of sensitivity to endocrine therapy in breast cancer [11]. Therefore, endocrine therapy is also commonly used in the treatment of recurrent ER+ ovarian cancer. However, no study has shown that ER positivity is related to the response to endocrine therapy in LGSOC, or ER+ ovarian cancer in general. Reported ORRs to endocrine therapy in LGSOC range between 9% and 14%, with aromatase inhibitors showing the highest ORRs [12,13]. Besides these minimal ORRs, endocrine therapy demonstrates a relatively high clinical benefit of around 60% in LGSOC [12,14].

In ER+ metastatic breast cancer, multiple mechanisms of resistance have been characterized. However, in ER+ ovarian cancer, these mechanisms remain largely unexplored. Given the similarities between ER+ breast cancer and ER+ ovarian cancer, we tried to uncover mechanisms of endocrine resistance in LGSOC. By extrapolating insights from these mechanisms observed in ER+ breast cancer, we can generate potential hypotheses of endocrine resistance in LGSOC that warrant further investigation. Moreover, this review aims to promote the exploration of innovative therapeutic approaches to overcome the challenge of endocrine resistance in rare ovarian cancer, where treatment options are currently limited.

## 2. Mechanisms of Estrogen Receptor Signaling

ER signaling is a complex process mediated by various estrogen receptor isoforms, including estrogen receptor alpha (ERα, *ESR1*) and estrogen receptor beta (ERβ; *ESR2*), both primarily functioning in the cell nucleus. Estrogen binds to the ER, resulting in receptor dimerization. The ligand-bound ER dimers translocate from the cytoplasm to the nucleus. In the nucleus, ER dimers bind to specific DNA sequences known as estrogen receptor elements (EREs), located in the promoter regions of the target genes. This ER-mediated transcriptional regulation leads to changes in the expression of target genes involved in various cellular processes, including cell proliferation, survival, differentiation, and metabolism. This classical genomic signaling is further enhanced via G protein-coupled estrogen receptor 1 (GPER1), a membrane-bound protein receptor that binds estrogen and activates multiple downstream signaling cascades such as the MAPK/ERK pathway and PI3K/AKT/mTOR pathway [15]. This so-called non-genomic signaling can elicit rapid cellular responses, including changes in cell proliferation, migration, and survival [16].

In ER+ breast cancer, estrogen signaling primarily involves the classical ER-mediated transcriptional activation of target genes leading to cancer cell proliferation and survival [16].

In ER+ ovarian cancer, the role of estrogen signaling is less well defined, and other pathways may be more influential in driving tumor growth and progression [17]. In ovarian cancer models, the role of ERα in the growth regulation of ovarian cancer cells has been shown in vitro and in vivo [15].

## 3. Possible Mechanisms of Endocrine Resistance

Although the majority of ER+ breast cancer initially responds to endocrine therapy, approximately 15% to 20% of tumors inherently show resistance to the therapy and an additional 30% to 40% develop resistance over time [18]. In LGSOC, endocrine therapy is often used for recurrent disease [12,13]. However, more than 85% of recurrent LGSOCs are not responsive to endocrine therapy, demonstrating endocrine resistance is an even greater challenge in these types of tumors.

### 3.1. ESR1 Alterations

Alterations in the *ESR1* gene can potentially reduce the responsiveness of LGSOC to endocrine therapy, especially aromatase inhibitors (e.g., letrozole, anastrozole, exemestane, etc.), as suggested by findings in breast and endometrial cancers [19,20,21,22,23,24,25].

#### 3.1.1. Loss of ER Expression

In some cases, breast cancer cells may lose their ER expression over time, resulting in the decreased efficacy of endocrine therapy [18,26]. The transcriptional repression of the *ESR1* gene might be caused by epigenetic modifications, including the abnormal CpG island methylation of the ER promoter and histone deacetylation mediated by histone deacetylase enzymes. These modifications result in a condensed nucleosome structure that restricts the transcription process [27]. In ovarian cancer, the loss of ER has not been documented yet.

#### 3.1.2. Mutations in the ESR1 Gene

Mutations in the ligand-binding domain of the *ESR1* gene (Figure 1) are observed in resistant breast cancer cells [21,28]. Activating *ESR1* mutations can contribute to tumor cell resistance through several mechanisms. In metastatic breast cancer, about 50% of endocrine-resistant cases are associated with an *ESR1* mutation. However, merely having an *ESR1* mutation is not enough to cause complete endocrine resistance [21,29].


*Ligand-independent activation*


The most common mutations in the *ESR1* gene are D538G and Y537S (Figure 1). They can alter the structure and function of the receptor, leading to reduced binding affinity to endocrine therapies and a decreased response to treatment. These activating mutations in the ligand-binding domain of *ESR1*, especially in helix 12, are an important mechanism of resistance to aromatase inhibitors [20,30]. Aromatase inhibitors block the aromatase enzyme and, therefore, inhibit the conversion of androgens into estrogens. This mechanism of action leads to a significant decrease in estrogen levels in the body. Normally, ERα requires the binding of estrogen to become activated. However, with an activating *ESR1* mutation, the receptor can be constitutively active even in the absence of estrogen. This means that the tumor cells can continue to grow and proliferate, despite the inhibition of estrogen production caused by aromatase inhibitors.

In recent publications, *ESR1* mutations have been detected in a subset of patients with LGSOC and uterine endometrioid carcinomas, highlighting the potential significance of these mutations in hormone-responsive gynecological malignancies. There are some case reports of *ESR1* mutations identified in LGSOC tumors. More specifically, McIntyre et al. detected an *ESR1* Y537S mutation in one patient with LGSOC when analyzing 26 primary tumor samples [19]. Additionally, Stover et al. also identified an *ESR1* Y537S mutation in a patient with LGSOC. This patient had a sustained response to endocrine therapy for 5 years but developed progressive disease with an isolated recurrent lesion that harbored an *ESR1* Y537S mutation [25]. Gaillard et al. reported the frequency of *ESR1* mutations in gynecological malignancies, which was relatively low overall (3.0% of all cases). Whether *ESR1* mutations are enriched in low-grade gynecological malignancies could not be determined because of the restricted information regarding tumor histology subtypes [24]. Fader et al. investigated *ESR1* Y537S mutations in three patients diagnosed with recurrent LGSOC, but no mutations were detected [31].

Stergiopoulou et al. evaluated the frequency and the clinical relevance of *ESR1* mutations in HGSOC. They reported the presence of *ESR1* mutations in 9 out of 60 (15%) FFPE samples and in 11 out of 80 (13.8%) circulating tumor DNA (ctDNA) samples from advanced and metastatic ovarian cancer patients. However, these samples included all types of ovarian cancer [32]. Besides this study and the case reports mentioned before, there has been a lack of subsequent reports regarding the identification of *ESR1* mutations in ovarian cancer. Therefore, further investigations are needed to explore the prevalence and clinical implications of *ESR1* mutations in LGSOC. These *ESR1* mutations can be monitored using ctDNA, making it a promising tool to predict endocrine resistance in these patients [33].


*Activation of alternative signaling pathways*


*ESR1* mutations can lead to the activation of downstream signaling pathways, like for example the PI3K/AKT/mTOR pathway, which can promote cell survival and growth. The activation of alternative pathways can compensate for the inhibition of estrogen signaling caused by endocrine therapy and render the tumor cells resistant to treatment [21]. Consequently, the effectiveness of endocrine therapy can be compromised when cancer cells harbor activating *ESR1* mutations. These mutations allow ER signaling pathways to remain active and alternative growth-promoting pathways to promote tumor growth. To overcome this resistance, strategies may involve targeting these modified signaling pathways and developing novel therapies that specifically target these *ESR1* mutations [21].

### 3.2. Crosstalk between ER and Growth Factor Signaling Pathways

As depicted below, breast cancer cells can activate alternative signaling pathways, such as growth factor receptor pathways (e.g., HER2, EGFR, and FGFR), MAPK/ERK pathway, and PI3K/AKT/mTOR pathway, due to alterations (mutations or amplifications) in multiple genes. Subsequently, this can activate alternative survival and cell proliferation signals and contribute to endocrine resistance (Figure 2).


*Growth factor receptor pathways*


Research has shown that the amplification of HER2 (ERBB2), a receptor tyrosine kinase (RTK), can lead to the activation of the alternative MAPK/ERK pathway and PI3K/AKT/mTOR survival pathways. Both intracellular pathways play an important role in gene expression regulation, cellular growth, motility, and survival [34].

ERBB2 amplification also occurs in LGSOC, with reported rates ranging from 1.5% to 11.5% [35,36]. The clinical significance of ERBB2 amplification in LGSOC is not yet fully understood. Anglesio and colleagues found that ERBB2-activating mutations are associated with an increase in the activity of the MAPK/ERK signaling pathway [36]. However, further research is necessary to better understand the role of ERBB2 amplification in LGSOC and to develop new treatment strategies.

Aberrations in components of the MAPK pathway, including *NF1*, *KRAS*/*NRAS*/*HRAS*, *BRAF*, and *MAP2K1*, are often reported in metastatic breast cancer [29,37]. Alterations in these genes may lead to increased or uncontrolled cell proliferation, leading to resistance to endocrine therapy in breast cancer [38]. As mentioned before, LGSOC is also molecularly characterized by MAPK pathway mutations. Several studies have reported *KRAS* mutations in 16–44% of cases, *BRAF* in 2–20%, and *NRAS* in as many as 26% of cases [39]. Further studies are necessary to investigate the correlation between these mutations and endocrine resistance in ER+ ovarian cancer.

The hyperactivation of the PI3K/AKT/mTOR pathway has also been associated with endocrine resistance in ER+ breast cancer [40,41,42]. A study performed by Beltrame et al. in ovarian cancer showed mTOR missense mutations in HGSOC and LGSOC [43]. However, it seems evident that LGSOC is characterized by a relatively low frequency of aberrations in the PI3K/AKT/mTOR pathway [44,45]. The phase II trial in patients with unresectable LGSOC comparing the combination of pimasertib, a MEK inhibitor, with SAR245409, a PI3K inhibitor, to pimasertib alone was terminated early due to high rates of discontinuation and low ORRs [46].

### 3.3. Epigenetic Modification

Endocrine resistance has previously been shown to be associated with epigenetic alterations in breast cancer, including DNA methylation, chromatin accessibility, histone modifications, and the binding of different transcription factors, due to their effect on gene expression [47,48]. Differential DNA methylation has been implicated in endocrine-resistant tumors [49]. For example, in breast cancer, different methylation profiles were found between tamoxifen-sensitive and tamoxifen-resistant cell lines [50]. Another study showed that the hypermethylation of estrogen-responsive enhancers modulates endocrine response in cell lines, which could be used to identify patients who positively respond to endocrine therapy [51]. Furthermore, the methylation profile of estrogen-responsive enhancers can increase during endocrine therapy and differ between endocrine-sensitive and endocrine-resistant patients [52].

In ER+ breast cancer, several preclinical studies and early-phase clinical trials have investigated the efficacy of epidrugs in combination with endocrine therapies or other targeted agents in ER+ breast cancer [47]. While some promising results have been observed, further research is needed to determine the optimal treatment combinations, patient selection criteria, and long-term outcomes.

MiRNAs have the capacity to regulate ER expression. In breast cancer, they play pivotal roles in both normal breast development and breast tumor formation [53]. Research states that miRNAs are dysregulated in endocrine-resistant breast cancer [53,54]. Several upregulated as well as downregulated miRNAs and their targets have been associated with endocrine resistance. These identified miRNAs can be useful as predictive serum biomarkers for developing endocrine resistance. In this way, patients who will or will not benefit from endocrine therapy can be determined.

In LGSOC, epigenetic research has not been explored yet.

## 4. Novel Potential Therapeutic Strategies

In the treatment of ER+ breast cancer, emerging endocrine therapies are being developed to overcome common mechanisms of endocrine resistance, for example, *ESR1* mutations, including combination therapies and next-generation selective estrogen receptor degraders (SERDs) and selective estrogen receptor modulators (SERMs), but also other new classes of endocrine therapies [55]. In this regard, it is important to understand that resistance to endocrine therapy can be agent-dependent. For instance, when aromatase inhibitor therapy has failed, tumors can still respond to alternative endocrine therapy approaches, including a different class of aromatase inhibitors (steroidal versus non-steroidal), SERMs (e.g., tamoxifen) or SERDs (e.g., fulvestrant) or other next-generation endocrine therapies [56].

### 4.1. Combination Therapies with Molecularly Targeted Agents

In ER+/HER2- breast cancer, the use of molecularly targeted therapies in combination with endocrine therapy has been widely explored. These compounds include cyclin-dependent kinase 4 and 6 (CDK4/6) inhibitors, mitogen-activated protein/extracellular signal-regulated kinase (MEK) inhibitors, and PI3K/mTOR inhibitors. The main goals are to improve patient outcomes by targeting multiple pathways and to reduce side effects and the development of resistance to therapy [57]. Following encouraging outcomes in clinical trials in ER+ breast cancer, distinct molecularly targeted combination agents have also been or are being tested in patients with LGSOC (Table 1) [58].

#### 4.1.1. CDK4/6 Inhibitors Combinations

In patients with advanced and metastatic ER+ breast cancer, the addition of CDK4/6 inhibitors to endocrine therapy has significantly increased survival rates [59,60,61]. Several trials have investigated the efficacy and safety of CDK4/6 inhibitors and they have been approved for the treatment of advanced ER+ and early high-risk breast cancer [60,61,62,63].

The results of these trials can form the rationale for designing clinical trials with the same drug combination in patients with ER+ ovarian cancer. Preclinical research demonstrated the biological activity of palbociclib in ovarian cancer cells with low p16 expression [64]. However, the use of CDK4/6 inhibitors as a single agent in a phase II clinical trial showed limited benefits in recurrent ovarian cancer with Rb-proficiency and low p16 expression, with a median progression-free survival (PFS) of 3.7 months [65,66]. In this trial, all types of epithelial ovarian tumors were included; no distinction was made between HGSOC and LGSOC. Nevertheless, combination therapies with CDK4/6 inhibitors seem to have potential in the treatment of LGSOC. Colon-Otero et al. found a promising clinical benefit of ribociclib in combination with letrozole in three out of three patients with recurrent LGSOC, including one complete response and two partial responses [67].

Upcoming and ongoing clinical trials testing the efficacy of CDK4/6 inhibitors in combination with endocrine therapy in LGSOC are listed in Table 1 [68].

#### 4.1.2. MEK Inhibitor Combinations

Previous research has shown that mitogen-activated extracellular signal-regulated kinase (MEK) inhibitors increase PFS and ORRs in patients with recurrent LGSOC [69], with ORRs ranging between 15% and 26%. However, there is some evidence that the addition of another therapy to MEK inhibitors, for example, endocrine therapy or targeted therapy, can improve the ORRs even more [70,71].

In advanced breast cancer, the addition of selumetinib (a MEK inhibitor) to fulvestrant resulted in a poor tolerance and worse disease control rate than with fulvestrant alone [72]. The authors hypothesize selumetinib may have deteriorated the effect of fulvestrant.

However, in an ovarian cancer mouse model, the addition of a MEK inhibitor to fulvestrant improved the tumor response compared with a MEK inhibitor alone [73]. Additionally, a case report of a heavily pretreated and endocrine- and platinum-resistant patient with LGSOC treated with trametinib and fulvestrant revealed a PFS of 9 months [71]. In an upcoming trial in patients with LGSOC, the combination of regorafenib, a multikinase inhibitor, with fulvestrant will be investigated to provide more insight into the efficacy of MAPK pathway inhibitors in combination with endocrine therapy (NCT05113368).

Additionally, there is some evidence that combination therapies with inhibitors of the MAPK pathway and inhibitors of the PI3K/AKT/mTOR pathway could provide benefits in LGSOC because they might have a synergistic activity [74]. However, additional studies investigating the efficacy of combined MEK and PI3K inhibition are necessary to evaluate its utility in LGSOC [46].

#### 4.1.3. PI3K/mTOR Inhibitors Combinations

For the treatment of advanced ER+ breast cancer, several clinical trials have evaluated combinations of endocrine therapy and inhibitors of the PI3K/AKT/mTOR pathway, including mTOR inhibitors and dual PI3K/mTOR inhibitors [75]. In the SOLAR-1 trial, treatment with the PI3Kα-specific inhibitor alpelisib in combination with fulvestrant improved outcomes in patients with PIK3CA-mutated ER+ advanced breast cancer [76].

The phase II basket study in ER+ recurrent, metastatic gynecological cancers, including LGSOC, which aims to evaluate the efficacy of letrozole in combination with alpelisib or ribociclib (ACTRN12621000639820), is an expansion upon the PARAGON trial, which focused on the use of single-agent anastrazole [13]. It aims to assess whether combining letrozole with alpelisib or letrozole with ribociclib results in a higher ORR. Participants will be allocated to one of the two treatment groups based on the PIK3CA mutation status.

In the BOUQUET trial, a study evaluating the efficacy and safety of multiple biomarker-driven therapies in patients with recurrent and advanced rare epithelial ovarian tumors, several combination therapies were introduced based on alterations in the tumor DNA. In Table 1, a detailed overview of the different therapies in the BOUQUET trial is defined. However, this trial was prematurely terminated due to unknown reasons.

Additional clinical trials on the combination of molecularly targeted agents with endocrine therapy are strongly warranted to investigate their synergistic effect in ER+ ovarian cancer. In the future, it is imperative for clinical trials to incorporate a broader array of biomarkers for patient selection. This proactive approach will significantly augment the overall efficacy and clinical benefit of the trial participants.

## 5. Next-Generation SERMs and SERDs to Overcome Endocrine Resistance

In ER+ breast cancer, next-generation oral SERMs and SERDs are increasingly being applied in clinical research, both as single agents and in combination with confirmed targeted therapies [38]. Additionally, other novel endocrine therapies, including proteolysis targeting chimeras (PROTACs), selective estrogen receptor covalent antagonists (SERCAs), and complete estrogen receptor antagonists (CERANs) are also emerging (Figure 3).

SERMs competitively bind the ER and exhibit tissue-dependent agonistic or antagonistic properties. In breast tumors, they potentiate an anti-estrogenic effect by downregulating the transcriptional activity of EREs. Tamoxifen is the most commonly used SERM in the treatment of ER+ metastatic breast cancer. However, recently, the development of novel agents has emerged [55].

SERDs also bind the ER and, therefore, inhibit receptor dimerization, causing the degradation and downregulation of the ER. Fulvestrant was the first-in-class approved SERD and is used to treat ER+ metastatic and advanced breast cancer alone, as well as in combination with CDK4/6 inhibitors. However, fulvestrant seems ineffective in patients with advanced breast cancer and *ESR1* mutations [77].

In recent years, multiple novel oral SERMs, SERDs, and other next-generation endocrine therapies have been investigated in clinical trials with ER+ breast cancer [78]. An overview of completed trials in advanced and metastatic ER+ breast cancer with new endocrine drug therapies to overcome endocrine resistance can be found in Table 2.

Overall, these new agents have more potent anti-estrogen activities and have properties to overcome endocrine resistance, including the degradation of the ER, higher potency and specificity, and the ability to target *ESR1* mutations. These properties could lead to a more effective and durable response in patients with ER+ breast cancer. These novel endocrine agents have shown activity after fulvestrant and CDK4/6 inhibitors, and on mutant *ESR1* in breast cancer. Hence, these agents also offer a viable possibility for determining a ‘what comes next’ approach in addressing LGSOC.

Phase II ELAINE demonstrated a numerically improved PFS and ORR with lasofoxifene compared to fulvestrant. These findings support further investigation of lasofoxifene in this patient population [79].

The phase III EMERALD trial demonstrated improved outcomes with the oral SERD elacestrant in ER+ metastatic breast cancer compared with standard-of-care endocrine therapy, including fulvestrant or aromatase inhibitors. These patients progressed on one or two prior lines of endocrine therapy and were pretreated with a CDK4/6 inhibitor [80]. Based on these positive results, elacestrant was granted accelerated FDA approval in November 2023 for the treatment of this patient population.

Among the *ESR1*-mutated population, a longer prior exposure to CDK4/6 inhibitors was associated with an extended PFS. Additionally, encouraging findings in other novel SERDs, including amcenestrant [81], camizestrant [84], giredestrant, imlunestrant, and rintodestrant, have led to the development of several clinical trials in advanced ER+ breast cancer.

Other novel endocrine therapies including SERMs (e.g., lasofoxifene) [79], PROTACs (e.g., ARV-471) [86], SERCAs (e.g., H3B-6545) [87], and CERANs (e.g., OP-1250) have emerged for the treatment of ER+ breast cancer Table 3.

PROTACs bind to the ER and recruit the E3 ubiquitin-ligase, leading to ER ubiquitination and the subsequent proteasomal degradation of the receptor [86]. SERCAs are a class of compounds that covalently bind to a unique cysteine residue at position 530. This cysteine residue is not present in other hormone receptors, leading to the inactivation of both wild-type and mutant ERs [87]. CERANs inhibit activation function 1 (AF1) and activation function 2 (AF2), both domains of the ER that activate gene transcription. CERANs can bind both wild-type and mutant ERs [88].

However, clinical trial data need to be collected further to draw any strong conclusions about the efficacy and safety of these next-generation anti-estrogens. An overview of ongoing trials in advanced and metastatic ER+ breast cancer with new endocrine drug combination therapies to overcome endocrine resistance can be found in Table 3. Ongoing randomized trials in advanced and metastatic ER+ breast cancer with new endocrine drug combination therapies to overcome endocrine resistance (randomized) and Table 4 (non-randomized).

In ovarian cancer, next-generation SERMs and SERDs are still largely unexplored, despite their potential clinical benefit in ER+ breast cancer. It would be of great interest to explore the activity of these agents in patients with LGSOC, especially when resistance to other endocrine therapeutic agents has been developed.

**Table 3 cancers-16-01862-t003:** Ongoing randomized trials in advanced and metastatic ER+ breast cancer with new endocrine drug combination therapies to overcome endocrine resistance.

Drug(s)	Mode of Action	Clinical Trial	Phase	Indication	Design	Primary Endpoint	Reference
Giredestrant combination therapy	SERD	persevERA NCT04546009	III	ER+HER2- advanced or metastatic breast cancer, first-line	Giredestrant combined with palbociclib vs. palbociclib + letrozole	PFS, as determined by the investigator according to RECIST v1.1	Turner et al. [89]
Camizestrant combination therapy	SERD	SERENA-4 NCT04711252	III	ER+/HER2- advanced or metastatic breast cancer, first-line	Camizestrant combined with palbociclib vs. palbociclib + aromatase inhibitor (anastrozole)	PFS, as determined by the investigator according to RECIST v1.1	Im et al. [90]
Giredestrant combination therapy	SERD	evERA NCT05306340	III	ER+/HER2- advanced or metastatic breast cancer, who received previous treatment with a CDK4/6i and ET	Giredestrant combined with everolimus vs. physician’s choice of endocrine therapy + everolimus	PFS, as determined by the investigator according to RECIST v1.1	Mayer et al. [91]
Giredestrant combination therapy	SERD	MORPHEUS NCT04802759	Ib/II	ER+/HER2- advanced or metastatic breast cancer, second- or third-line	Giredestrant alone or combined with abemaciclib, palbociclib, ribociclib, ipatasertib, inavolisib, everolimus, or samuraciclib	ORR, as determined by the investigator according to RECIST v1.1	Oliveira et al. [92]
Camizestrant combination therapy	SERD	SERENA-6 NCT04964934	III	ER+/HER2-, *ESR1* mutated, advanced or metastatic breast cancer on first-line aromatase inhibitor + CDK4/6i before disease progression	Switch to camizestrant combined with CDK4/6 inhibitor vs. continue current treatment with aromatase inhibitor + CDK4/6 inhibitor	PFS, as determined by the investigator according to RECIST v1.1	Turner et al. [93]
Imlunestrant monotherapy or combination therapy	SERD	EMBER-3 NCT04975308	III	ER+/HER2- advanced or metastatic breast cancer, who received previous treatment with ET, +/− CDKi	Imlunestrant monotherapy vs. combined with abemaciclib vs. investigator’s choice of endocrine therapy	PFS, as determined by the investigator according to RECIST v1.1	Jhaveri et al. [94]

Abbreviations: CDKi, cyclin-dependent kinase inhibitor; ER+, estrogen receptor-positive; HER2-, human epidermal growth factor receptor 2-negative; ET, endocrine therapy; PFS, progression-free survival; ORR, overall response rate; SERD, selective estrogen receptor degrader; SERM; selective estrogen receptor modulator; RECIST v1.1, Response Evaluation Criteria in Solid Tumors version 1.1.

**Table 4 cancers-16-01862-t004:** Ongoing non-randomized trials in advanced and metastatic ER+ breast cancer with new endocrine drug (combination) therapies to overcome endocrine resistance.

Drug(s)	Mode of Action	Clinical Trial	Phase	Indication	Design	Primary Outcome	Reference
Rintodestrant combination therapy	SERD	NCT03455270	I	ER+/HER2- advanced or metastatic breast cancer, 2nd-line or more	Rintodestrant monotherapy vs. combined with palbociclib	Dose-limiting toxicity Recommended phase 2 dose	Adreano et al. [95]
ZN-c5 combination therapy	SERD	NCT04514159	Ib	ER+/HER2- advanced or metastatic breast cancer, 2nd-line or more, no prior CDK4/6i	ZN-c5 combined with abemaciclib	Maximum tolerated dose Recommended phase 2 dose	Keogh et al. [96]
ZN-c5 monotherapy or combination therapy	SERD	NCT03560531	I/II	ER+/HER2- advanced or metastatic breast cancer, 2nd-line or more	ZN-c5 monotherapy or combined with palbociclib	Maximum tolerated dose Recommended phase 2 dose	Abramson et al. [97]
Borestrant monotherapy and combination therapy	SERD	ENZENO NCT04669587	I/II	ER+/HER2- advanced or metastatic breast cancer at any line	Borestrant monotherapy and combined with palbociclib	Safety and tolerability	NA
D-0502 monotherapy and combination therapy	SERD	NCT03471663	I	ER+/HER2- advanced or metastatic breast cancer, 2nd-line or more	D-0502 monotherapy and combined with palbociclib	Safety and tolerability	Osborne et al. [98]
ARV-471 monotherapy and combination therapy	PROTAC	NCT04072952	I/II	ER+/HER2- advanced or metastatic breast cancer, 2nd-line or more	ARV-471 monotherapy and combined with palbociclib	Safety and tolerability	Hamilton et al. [86]
OP-1250 monotherapy	CERAN	NCT04505826	I/II	ER+/HER2- advanced or metastatic breast cancer, 2nd-line or more	OP-1250 monotherapy	Safety and tolerability	Hamilton et al. [88]
H3B-6545 combination therapy	SERCA	NCT04288089	Ib	ER+/HER2- advanced or metastatic breast cancer, 3rd-line or more	H3b-6545 combined with palbociclib	Maximum tolerated dose Recommended phase 2 dose	Johnston et al. [87]

Abbreviations: CERAN, complete estrogen receptor antagonist; ER+, estrogen receptor-positive; HER2-, human epidermal growth factor receptor 2-negative; PROTAC, proteolysis targeting chimera; SERCA; selective estrogen receptor covalent antagonist; SERD, selective estrogen receptor degrader; SERM, selective estrogen receptor modulator.

## 6. Future Directions

Initiating clinical trials to investigate the efficacy of novel endocrine combination therapies, previously shown successful in breast cancer, is one of the most important goals of future research on low-grade serous ovarian cancer. Especially in patients already treated with traditional endocrine therapies who developed endocrine resistance to this treatment. Next-generation endocrine therapeutic agents, designed to specifically target mechanisms of endocrine resistance, are emerging in the field of breast cancer and represent a pivotal area of research in ER+ ovarian cancer.

Patient-derived ovarian cancer organoid models could provide a powerful tool to screen novel endocrine combination therapies in the future [99,100]. These models present a promising platform for preclinical drug testing and screening, accelerating the discovery of effective treatments and exploring the repurposing of existing medicines for new indications, including rare cancers. Ultimately, this may lead to the development of tailored therapeutic approaches and improved outcomes for patients.

## 7. Conclusions

Research on the mechanisms of endocrine resistance in ER+ ovarian cancer is currently limited. Although there may be some overlap in the mechanisms of endocrine resistance between ER+ breast cancer and ER+ ovarian cancer, there are also distinct differences. As a result, findings from studies on ER+ breast cancer cannot be directly applied to ER+ ovarian cancer, highlighting the need for further research. Therefore, it is crucial to develop translational research models to identify biomarkers that can predict response or resistance to endocrine (combination) therapies and to define novel treatment strategies.

In ER+ breast cancer, the utilization of endocrine combination therapies has demonstrated enhanced efficacy, resulting in improved control of breast cancer and lowered rates of recurrence. Ongoing research endeavors aim to refine these endocrine combination therapies for ER+ breast cancer, particularly in cases of endocrine-resistant disease. This involves assessing novel agents such as next-generation SERDs and SERMs, as well as investigating alternative combinations.

Exploring next-generation SERDs and SERMs in the context of ER+ ovarian cancer presents a promising area of research. While current clinical data regarding their application in ER+ ovarian cancer remain scarce, the ongoing research and clinical trials hold promise for the emergence of innovative and efficacious treatment strategies for this complex disease.

## Figures and Tables

**Figure 1 cancers-16-01862-f001:**
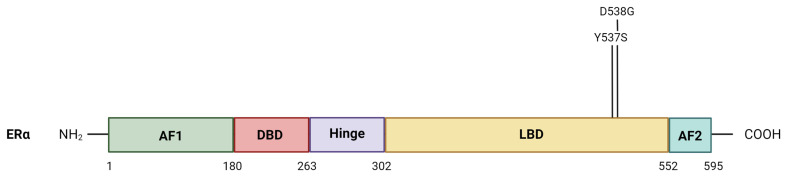
Schematic representation of the estrogen receptor-α (ERα), including the most common LBD point mutations. The structural domains of ERα are shown, including the transcription activation function domains (AF1 and AF2), the DNA-binding domain, the receptor dimerization and nuclear localization (hinge) domain, and the ligand-binding domain. Abbreviations: AF1, activation function 1 domain; AF2, activation function domain 2; DBD, DNA-binding domain; LBD, the ligand-binding domain. Created with BioRender.com.

**Figure 2 cancers-16-01862-f002:**
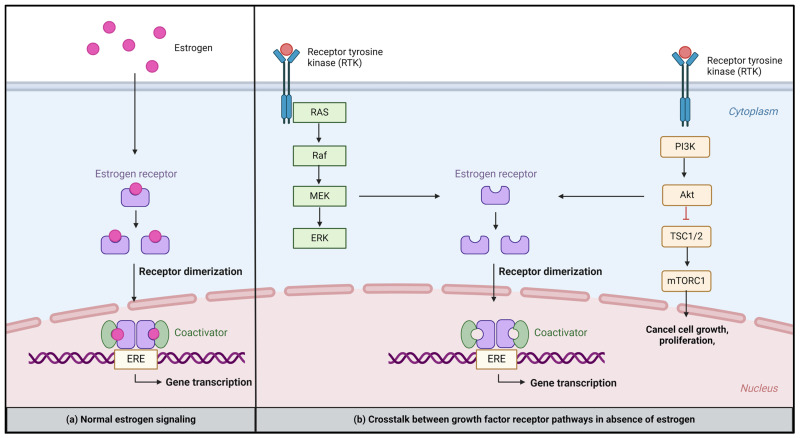
Activation of alternative pathways. (**a**) Normal estrogen signaling; (**b**) crosstalk between growth factor receptor pathways (MEK and PI3K/mTOR pathways) and the estrogen pathway. The estrogen pathway is activated in the absence of estrogen as a mechanism of endocrine resistance. Created with BioRender.com. Abbreviation: ERE; estrogen response element.

**Figure 3 cancers-16-01862-f003:**
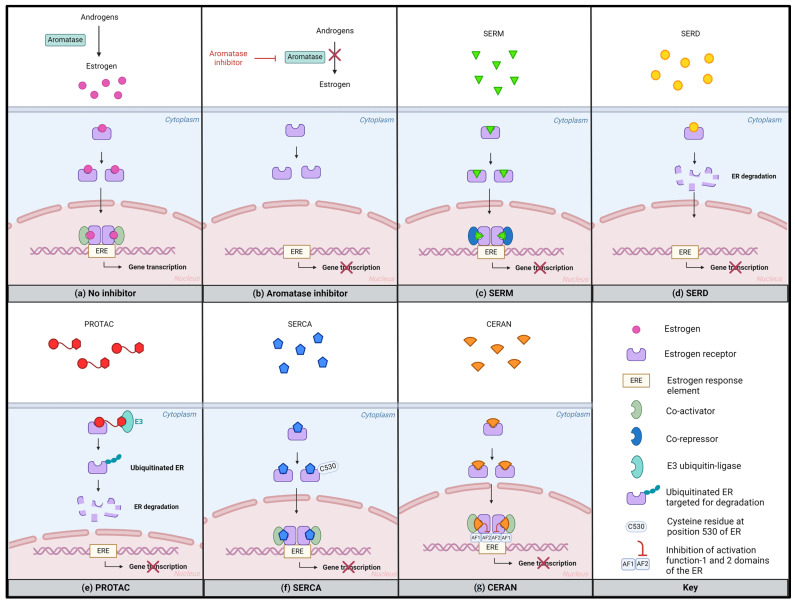
Mechanisms of action of different endocrine therapies. (**a**) Estrogens bind to the ER, promoting receptor dimerization and translocation to the nucleus. The estrogen-bound ER dimer regulates gene transcription, resulting in gene expression and cancer cell growth and survival; (**b**) aromatase inhibitors block the aromatase enzyme, preventing the conversion from androgens into estrogen and leading to the inhibition of gene transcription; (**c**) SERMs competitively bind to the ER, leading to the inhibition of ER-dependent gene transcription by recruiting transcriptional co-repressors; (**d**) SERDs bind to the ER, inhibiting its translocation to the nucleus, and promoting its destabilization and degradation; (**e**) PROTACs mediate an interaction between the ER and the E3 ubiquitin-ligase complex, facilitating ubiquitination of the ER and subsequently its degradation; (**f**) SERCAs covalently bind to the C530 residue of both wild-type and mutant ERs, leading to the inactivation of these receptors; (**g**) CERANs bind the ER, leading to the inactivation of AF1 and AF2 of both wild-type and mutant ERs, consequently inhibiting gene transcription. Created with BioRender.com. Abbreviations: AF1, activation function 1; AF2, activation function 2; CERAN, complete estrogen receptor antagonists; ER, estrogen receptor; ERE, estrogen response element; PROTAC, proteolysis targeting chimera; SERCA, selective estrogen receptor covalent antagonist; SERD, selective estrogen receptor modulator; SERM, selective estrogen receptor modulator.

**Table 1 cancers-16-01862-t001:** Ongoing and upcoming clinical trials with combination therapies for LGSOC.

Drug	Mode of Action	Status	Phase	Indication	Biomarkers for Patient Selection	Enrolment	Clinicaltrials.gov
Ribociclib Letrozole	CDK4/6 inhibitor Aromatase inhibitor	Active, not recruiting	II	Recurrent LGSOC	NA	51 (actual)	NCT03673124
Abemaciclib Fulvestrant	CDK4/6 inhibitor SERD	Active, not recruiting	II	Advanced LGSOC, first-line, neo-adjuvant treatment	NA	18 (actual)	NCT03531645
Abemaciclib Letrozole	CDK4/6 inhibitor Aromatase inhibitor	Recruiting	II	Recurrent LGSOC	ER positivity	100 (estimated)	NCT05872204
Palbociclib Binimetinib	CDK4/6 inhibitor MEK inhibitor	Recruiting	II	*RAS*-mutant cancers, including LGSOC	KRAS/NRAS/HRAS or BRAF alterations, RAF mutations or RAF fusions	199 (estimated)	NCT05554367
Avutometinib (VS-6766) +/− Defactinib (VS-6063)	Dual RAF/MEK inhibitor FAK inhibitor	Recruiting	II	Recurrent LGSOC with or without a *KRAS* mutation	KRAS wild-type KRAS mutation	225 (estimated)	NCT04625270
Avutometinib (VS-6766) + Defactinib (VS-6063)	Dual RAF/MEK inhibitor FAK inhibitor	Recruiting	III	Recurrent LGSOC	KRAS wild-type KRAS mutation	270 (estimated)	NCT06072781
Pimasertib Voxtalisib (SAR245409)	MEK1/2 inhibitor PI3K/mTOR inhibitor	Completed	II	Recurrent borderline/low malignant potential and LGSOC	NA	65 (actual)	NCT01936363 [46]
Regorafenib Fulvestrant	Multikinase inhibitor SERD	Recruiting	II	Recurrent LGSOC	NA	31 (estimated)	NCT05113368
Letrozole Alpelisib OR Letrozole Ribociclib based on presence of PI3KCA mutation status	Aromatase inhibitor PI3Ka-inhibitor OR Aromatase inhibitor CDK4/6 inhibitor	Recruiting	II	Advanced gynecological cancers that express hormone receptors	ER and/or PR positivity	100 (actual)	ACTRN12621000639820
Biomarker-driven therapy *	AKT inhibitor MEK inhibitor Chemotherapy CDK4/6 inhibitor PI3K α inhibitor SERD Aromatase inhibitor VEGF-A inhibitor PARP inhibitor PD-L1 inhibitor Anti-HER2 antibody + cytotoxic agent	Prematurely terminated	II	Recurrent rare epithelial ovarian tumors	See footnote of this table	500 (estimated)	NCT04931342

* BOUQUET trial: *PIK3CA*/*AKT1*/*PTEN*-altered tumors => Ipatasertib (AKT inhibitor) and Paclitaxel (chemotherapy); *BRAF*/*NRAS*/*KRAS*/*NF1*-altered tumors => Cobimetinib (MEK inhibitor); *ERBB2*-amplified/mutant tumors => Trastuzumab Emtansine (anti-HER2 antibody + cytotoxic agent); no specific mutation => Atezolizumab (PD-L1 inhibitor) with Bevacizumab (VEGF-A inhibitor); ER+ tumors => Giredestrant (SERD) + Abemaciclib (CDK4/6 inhibitor) + LHRH agonist if peri- or pre-menopausal; *PIK3CA*-altered tumors => Inavolisib (PI3K-α inhibitor) + Palbociclib (CDK4/6 inhibitor); ER+ and *PIK3CA*-altered tumors => Inavolisib + Palbociclib + Letrozole (aromatase inhibitor); no specific mutation => Inavolisib + Olaparib (PARP inhibitor); ER+ and *PIK3CA*-altered tumors => Inavolisib + Giredestrant; *PIK3CA*-altered tumors => Inavolisib + Bevacizumab; no specific mutation => Atezolizumab + Bevacizumab + Cyclophosphamide (chemotherapy). Abbreviations: AKT, protein kinase B; CDK4/6, cyclin-dependent kinase 4 and 6; ER, estrogen receptor; FAK, focal adhesion kinase; HER2, human epidermal growth factor receptor 2; mTOR, mammalian target of rapamycin; NA, not applicable; PARP, poly ADP-ribose polymerase; PD-L1, programmed death-ligand 1; PI3K, phosphoinositide 3-kinase; PR, progesterone receptor; SERD, selective estrogen receptor degrader; RAF, rapidly accelerated fibrosarcoma; VEGF-A, vascular epidermal growth factor A.

**Table 2 cancers-16-01862-t002:** Completed trials in advanced and metastatic ER+ breast cancer with new endocrine drug therapies to overcome endocrine resistance.

Drug(s)	Mode of Action	Clinical Trial	Phase	Indication	Treatment Arms	Outcome Primary Endpoints	Reference
Lasofoxifene	SERM	ELAINE I NCT03781063	II	ER+/HER2- advanced or metastatic breast cancer with an *ESR1* mutation	Lasofoxifene vs. fulvestrant	Median PFS: 5.6 months (lasofoxifene) vs. 3.7 months (fulvestrant); *p* = 0.138 ORR was higher for lasofoxifene (14.3%) compared to fulvestrant (3.9%)	Goetz et al. [79]
Elacestrant	SERD/ SERM	EMERALD NCT03778931	III	ER+/HER2- advanced or metastatic breast cancer with 1–2 previous lines of endocrine therapy, including a CDK4/6 inhibitor	Elacestrant vs. standard endocrine therapy (fulvestrant/AI)	Median PFS: 2.8 months (elacestrant) vs. 1.9 months (standard endocrine therapy); HR: 0.664; *p* = 0.0019	Bidard et al. [80]
Amcenestrant	SERD	AMEERA-3 NCT04059484	II	ER+/HER2- advanced or metastatic breast cancer that progressed after prior treatment with an AI and a CDK4/6 inhibitor	Amcenestrant vs. physician’s choice of endocrine treatment	Median PFS: 3.6 months (amcenestrant) vs. 3.7 months (physician’s choice of endocrine treatment); HR: 1.05, *p* = 0.64	Tolaney et al. [81]
Giredestrant	SERD	acelERA NCT04576455	II	ER+/HER2- advanced or metastatic breast cancer that received 1 or 2 prior lines of systemic therapy	Giredestrant vs. physician’s choice of endocrine treatment	Median PFS: 5.6 months (giredestrant) vs. 5.4 months (physician’s choice of endocrine treatment); HR 0.81, *p* = 0.18	Martin et al. Martin Jimenez et al. [82,83]
Camizestrant	SERD	SERENA-2 NCT04214288	II	ER+/HER2- advanced or metastatic breast cancer	Camizestrant 75, 150 or 300 mg vs. Fulvestrant	Median PFS: 7.2 months (camizestrant 75 mg) vs. 3.7 (fulvestrant); *p* = 0.0124 Median PFS: 7.7 months (camizestrant 150 mg) vs. 3.7 (fulvestrant); *p* = 0.0161	Lawson et al. Oliveira et al. [84,85]

Abbreviations: ER+, estrogen receptor-positive; HER2-, human epidermal growth factor receptor 2-negative; HR, hazard ratio; PFS, progression-free survival; OS, overall survival; SERD, selective estrogen receptor degrader; SERM; selective estrogen receptor modulator.

## Data Availability

All primary data cited in this review are published as noted in the references section and available in the public domain.

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
