# Peer review of "Novel Endocrine Therapeutic Opportunities for Estrogen Receptor-Positive Ovarian Cancer—What Can We Learn from Breast Cancer?"

_cancers, 2024, doi:10.3390/cancers16101862_

Round 1

Reviewer 1 Report

Comments and Suggestions for Authors

In this review, Ottenbourgs and Van Nieuwenhuysen summarize the potential mechanisms of endocrine resistance in estrogen receptor-positive (ER+) breast cancer, which shares similar characteristics with low-grade serous ovarian cancer (LGSOC). They also explore potential therapeutic approaches to endocrine resistance in recurrent LGSOC, which poses significant clinical challenges with no available treatments.

Overall, it is well written and provides potential treatment strategies for patients with recurrent LGSOC. Minor comments for Table 1 and consistent formatting are listed below.

1. Table 1: It would be better to include the biomarkers used for patient selection, if applicable.

2. Please use abbreviations once they have been defined, some examples are listed below but not limited to, e.g. low-grade serous ovarian cancer to LGSOC (lines 20, 28, 30, 34, 103, 135......etc), estrogen receptor to ER (lines 62, 64, 357, 362), objective response rate to ORR (line 286), progression-free survival to PFS (lines 287, 395).

3. Line 258: mitogen-activated protein kinase was defined as MEK, should it be MAPK or did the authors want to say mitogen-activated extracellular signal-regulated kinase (MEK)?

4. Line 310: It is not clear which references [72] and/or (64) were used for PI3K/mTOR inhibitors.

Author Response

We kindly want to thank you for taking the time to comprehensively review our manuscript. Please find the detailed responses below and the corresponding revisions and corrections in track changes in the re-submitted files.

  • Comment 1: Table 1: It would be better to include the biomarkers used for patient selection, if applicable.

Response 1: We agree that this is interesting information to be added. Therefore, we added a row with biomarkers for patient selection in Table 1.

  • Comment 2: Please use abbreviations once they have been defined, some examples are listed below but not limited to, e.g. low-grade serous ovarian cancer to LGSOC (lines 20, 28, 30, 34, 103, 135......etc), estrogen receptor to ER (lines 62, 64, 357, 362), objective response rate to ORR (line 286), progression-free survival to PFS (lines 287, 395).

Response 2: Thank you for pointing this out. The abbreviations were adapted. Once they have been defined in the manuscript, they are abbreviated. 

  • Comment 3: Line 258: mitogen-activated protein kinase was defined as MEK, should it be MAPK or did the authors want to say mitogen-activated extracellular signal-regulated kinase (MEK)?

Response 3: We indeed meant the mitogen-activated protein/extracellular signal-regulated kinase kinase (MEK) inhibitor. This was adapted. 

  • Comment 4: Line 310: It is not clear which references [72] and/or (64) were used for PI3K/mTOR inhibitors.

Response 4: Indeed, reference [72] was used for PI3K/mTOR inhibitors. Consequently, reference (64) was deleted. This reference was wrongly inserted.

Additional clarifications:

Furthermore, a paragraph called 'Mechanisms of estrogen receptor signaling' was added on page 3 to elaborate a bit more on the mechanistic knowledge on the estrogen receptor.

Reviewer 2 Report

Comments and Suggestions for Authors

This authors aimed to highlight the underlying molecular mechanisms possibly driving endocrine resistance in LGSOC, while also exploring the available therapeutic opportunities to overcome this resistance. I consider the paper to be an interesting perspective. It may also have a wealth of clinical implications.

Author Response

Thank you very much for taking the time to review this manuscript.

Please find the revisions and corrections in track changes in the re-submitted files.

Reviewer 3 Report

Comments and Suggestions for Authors

In the present manuscript the Authors was to stimulate novel anti-estrogen therapeutic approaches for low-grade serous ovarian cancers (LGSOCs expressing the estrogen receptor α. Therapies with anti-estrogen drug gave good results in selected LGSOC patients but so far poor approach for patient selection or knowledge about resistance to these therapies in these tumors are reported.  Their approach to increase this knowledge was to extensively report about what it is known on estrogen receptor resistance in breast cancer. The title is interesting and the approach may be useful.

While the reported bibliography is exhaustive and the writing is very clear, the reader can get really the focus since it is so much reported about breast cancer and much less about LGSOC. Additionally, the title also contributes to make such confusion.

The narrative completely lacks the most mechanistic knowledge on the estrogen receptor and ovarian cancer. Perhaps adding this knowledge obtained in preclinical models in both ovarian and breast carcinomas will better introduce the resistance to hormone therapy, the new perspectives and trial.

Comments on the Quality of English Language

Minor editing is required.

Author Response

Thank you very much for taking the time to review our manuscript. Please find the detailed responses below and the corresponding revisions and corrections in track changes in the re-submitted files.

  • Comment 1: While the reported bibliography is exhaustive and the writing is very clear, the reader can get really the focus since it is so much reported about breast cancer and much less about LGSOC. Additionally, the title also contributes to make such confusion.

Response 1: We agree with this comment. Therefore, in the revised manuscript, we tried to provide further elaboration on LGSOC. Nonetheless, due to its rarity in comparison to ER+ breast cancer, there remains a significant gap in understanding. Through this review, we also aim to underscore the urgent need for additional research in LGSOC.

  • Comment 2: The narrative completely lacks the most mechanistic knowledge on the estrogen receptor and ovarian cancer. Perhaps adding this knowledge obtained in preclinical models in both ovarian and breast carcinomas will better introduce the resistance to hormone therapy, the new perspectives and trial.

Response 2: Unfortunately, our understanding of the mechanism of the estrogen receptor in ovarian cancer, particularly in LGSOC, remains limited. Despite this, we endeavored to outline the most significant general mechanisms currently known in a paragraph called 'Mechanisms of estrogen receptor signaling' on page 3 of the manuscript.

Furthermore, we have made some revisions to the English language.